# Imidacloprid Pesticide Causes Unexpectedly Severe Bioelement Deficiencies and Imbalance in Honey Bees Even at Sublethal Doses

**DOI:** 10.3390/ani13040615

**Published:** 2023-02-09

**Authors:** Jerzy Paleolog, Jerzy Wilde, Marek Gancarz, Dariusz Wiącek, Agnieszka Nawrocka, Aneta Strachecka

**Affiliations:** 1Department of Invertebrate Ecophysiology and Experimental Biology, University of Life Sciences in Lublin, Doświadczalna 50a, 20-280 Lublin, Poland; 2Department of Poultry Science and Apiculture, Faculty of Animal Bioengineering, Warmia and Mazury University in Olsztyn, Słoneczna 48, 10-957 Olsztyn, Poland; 3Institute of Agrophysics, Polish Academy of Sciences, Doświadczalna 4, 20-290 Lublin, Poland; 4Faculty of Production and Power Engineering, University of Agriculture in Kraków, Balicka 116B, 30-149 Kraków, Poland

**Keywords:** *Apis mellifera*, neonicotinoid, pesticide, biominerals, nutrition, bee healthcare

## Abstract

**Simple Summary:**

Honey bees deliver many valuable products and pollinate many crucial crops, and pesticides impair honey bee health in many ways. It is a serious problem of the contemporary agroecosystem, but the mechanisms of these phenomena have not been sufficiently explored. Therefore, we investigated the impact of the imidacloprid pesticide (IMD) on the bioelement body content in honey bees in field experiments using both sublethal considered field-relevant (5 ppb), and adverse (200 ppb) doses, which has not been studied before. Our findings revealed for the first time that IMD causes unexpectedly severe bioelement deficiencies in 69% of bioelements (32 assayed) and disturbs the balance between the levels of the remaining ones even at sublethal doses. The increase in three toxic bioelements was alarming. Consequently, we have suggested a new physiological mechanism regarding how nicotinoids may interfere with honey bee health status. This also sets out new directions for further research, pointing to the bioelement supplementation of the diet as an important element of honey bee preventive health care when the bee farms or amateur apiaries are located in an agrocenosis exposed to pesticides.

**Abstract:**

Pesticides impair honeybee health in many ways. Imidacloprid (IMD) is a pesticide used worldwide. No information exists on how IMD impact the bees’ body bioelement balance, which is essential for bee health. We hypothesized that IMD disturbs this balance and fed the bees (in field conditions) with diets containing 0 ppb (control), 5 ppb (sublethal considered field-relevant), and 200 ppb (adverse) doses of IMD. IMD severely reduced the levels of K, Na, Ca, and Mg (electrolytic) and of Fe, Mo, Mn, Co, Cu, Ni, Se, and Zn, while those of Sn, V, and Cr (enzymatic) were increased. Levels of P, S, Ti, Al, Li, and Sr were also decreased, while only the B content (physiologically essential) was increased. The increase in Tl, Pb, and As levels (toxic) was alarming. Generally, IMD, even in sublethal doses, unexpectedly led to severe bioelement malnutrition in 69% of bioelements and to a stoichiometric mismatch in the remaining ones. This points to the IMD-dependent bioelement disturbance as another, yet unaccounted for, essential metabolic element which can interfere with apian health. Consequently, there is a need for developing methods of bioelement supplementation of the honey bee diet for better preventing bee colony decline and protecting apian health status when faced with pesticides.

## 1. Introduction

Honey bees (*Apis mellifera* L.) are economically important as they deliver a wide range of products and by-products necessary for human health and diet, but first of all, they are pollinators of many crucial crops [1]. Nowadays a decline in honey bee populations accompanied by a worldwide deterioration in bee health is commonly observed in commercial, semi-commercial, and amateur beekeeping. One of the most important factors blamed for the decline of honey bee colonies is the exposure to pesticides [2]. Therefore, their effects on colony survival, health and behavior are studied all over the world [3,4]. Among the pesticides, imidacloprid [1-(6-chloro-3-pyridylmethyl)-2nitroimino-imidazolidine] is a neonicotinoid neuro-active systemic insecticide widely used in pest control, seed treatment, and fighting termites, fleas, etc. Consequently, it influences non-target honey bees, posing a threat for honey production, crop pollination, and plant biodiversity in natural ecosystems, particularly in the temperate and subtropical zones [1,5]. The nicotinoid pesticides, including imidacloprid (IMD), are blamed, among others, for the honey bee colony depopulation syndrome. It is important that even sublethal doses of neonicotinoids handicap longevity, reproduction, health status, cognition, and social behavior, making honey bee colonies more sensitive to pathogens and xenobiotics [6,7,8,9]. That is because IMD deteriorates immunity, detoxification, proteolytic abilities, antioxidative barriers, DNA methylation, carbohydrate and lipid metabolism, and many other important physiological mechanisms and pathways in honey bees [6,7,10,11,12,13,14,15]. However, there is a gap in this knowledge since, as far as we know, there is no evidence of the impact of neonicotinoids on bee body bioelement levels/balance, which, on the other hand, is essential for bee metabolism, and recent detailed review papers do not mention this [3,4,5,16]. Filling this gap is important for a better understanding of the mechanisms of the worldwide honey bee decline.

Previous studies of bioelement content in the bee body have been focused on using honey bees as bioindicators of environmental pollution and on determining the contamination of honey bee products [17,18]. However, studies concerning the function of bioelements and their impact on honey bee physiology are limited for the time being [16]. Nevertheless, bioelements are suggested to be essential for the functioning of most of the honey bee metabolic pathways [19,20], particularly those involved in bee development, overwintering, and pathogen resistance [21,22,23,24]. Since, as opposed to organic compounds, the bioelements are not converted to other elements during the metabolism, it is the proper balance between their tissue levels that is essential for organism fitness and health.

Assuming that IMD harms honey bees by disrupting their physiological processes in almost every possible way [4] and that our knowledge about the metabolic background of these phenomena is still limited, as well as assuming that the scarcity or excess of the majority of bioelements impairs a wide range of physiological functions in honey bees [19], it is surprising that there is no information available on the impact of IMD on the bioelement balance or deficits in honey bees. Therefore, we believe that for better understanding the honey bee colony depopulation and decline when the insects are faced with neonicotinoid pesticides in the contemporary agroecosystems, it is necessary to learn whether and how IMD may cause bioelement scarcity or excess in honey bee bodies. Furthermore, we hypothesize that IMD disrupts the balance between biominerals even in sublethal doses and consequently may handicap honey bee physiology and ultimately their health status. Hence, for the first time, we examined the influence of both the field-relevant, sublethal 5 ppb and adverse 200 ppb doses of IMD on the bioelement content in the entire bodies of worker bees. In other words, the 5 ppb level was similar to the level which was found in the field where IMD was used and was considered “field-relevant” or “field realistic” by other authors (compare [2,5,9]).

The aim of this research was to fill the gap in the knowledge about the mechanisms of IMD harmfulness to honey bees considering the bioelement balance in their bodies. This is also useful for developing methods of protecting honey bees and preventing their exposure to pesticides, particularly in sublethal doses which occur in the contemporary agroecosystems. Consequently, this may lead to the development of a new recommendation for bee health prophylaxis.

## 2. Materials and Methods

### 2.1. Field Experiment

The field part of this study was carried out in 2020 at the bee yard of the Warmia and Mazury University in Olsztyn, Poland (19.53 E, 53.50 N). Procedures described by [14,15] were applied as follows. The mini-plus hives (six frames of 251 × 159 mm placed in one super) were populated with *Apis mellifera carnica* worker bees to establish experimental colonies of similar strength and structure. The worker bees originated from a few source colonies belonging to the same commercial stock and both the experimental and the source colonies had been maintained in the same location for 2 years. Furthermore, the experimental colonies were headed by egg laying, one-year-old sister queens. Each of them was instrumentally inseminated with the semen of drones from the same mother-queen. Then, these experimental colonies, which were genetically and structurally similar and homogeneous, were used to set up three feeding groups. Each group consisted of 12 colonies. Bees in the control group were fed with pure sugar-water syrup (5:3 *w*/*w*). In the IMD5 group, the syrup contained 5 ppb of IMD (Bayer Health Care AG, Leverkusen, Germany) and the IMD200 group contained 200 ppb of IMD. The colonies were also supplied with commercial bee food (API-Fortune HF 1575, Bollène, France) with the addition of ground, mingled, multifloral pollen containing 0 ppb, 5 ppb, and 200 ppb of IMD in the control, IMD5, and IMD200 groups, respectively. In this case the pollen was a unified source of biominerals. Importantly, the natural bee food containing potential bioelement resources was not available at that location during the study, and IMD was not used in the region. Consequently, the only source of IMD for the experimental colonies was the syrup and the commercial bee food, and the bioelement contents of diets of the experimental colonies ought to be similar to each other.

After 6 weeks of such feeding, a few dozen worker bees (hereafter called a “hive sample”) were sampled from the central combs from each colony within each feeding group (control, IMD5, and IMD200) for further laboratory analyses performed at the Institute of Agrophysics, Polish Academy of Sciences, Lublin, Poland. These workers were not foragers and must have developed (preimaginal and imaginal stages) during the chronic exposure to IMD, which lasted 6 weeks. The workers from each of these hive samples were kept for 1 day in a separate cage, were supplied with the sugar candy (without IMD and pollen), and subsequently transported to Lublin, which lasted about 12 h. Such a method for the collection of bee samples, together with this procedure and schedule of the bee transport, prevented the presence of nectar and pollen foraged in the field in guts of the sampled bees.

### 2.2. Laboratory Assays

Fifteen worker bees, showing the proper activity and mobility were selected from each hive sample and frozen (−25 °C) to constitute a pooled sample for the hive. The entire bodies of the bees were then analyzed. The content of individual bioelements in each of the samples was assayed using inductively coupled plasma optical emission spectrometry (ICP-OES, iCAP Series 6500, Thermo Scientific, Waltham, MA, USA). The sample mineralization was performed in a Microwave Digestion System (Bergh of Speedwave, Eningen, Germany) applying optical, temperature, and pressure monitoring of each sample at the time of the acid digestion in Teflon vials (type DAP 100). The mineralized worker bee bodies were digested with 7 mL HNO_3_ (65% *v*/*v*) and 3 mL H_2_O_2_ (30% *v*/*v*). Each of the samples was performed in three repetitions. The average of these three measurements was taken as the value of the hive pooled sample. Therefore, the data base was 3 × 12 hive pooled samples with 15 worker bees in each (three measurement replicates in each).

The mineralization procedure details (compare [24] as follows: 15 min from room temperature to 140 °C, 5 min at 140 °C, 15 min from 140 °C to 185 °C, 10 min at 185 °C, and then cooling down to room temperature. The pressure did not exceed 20 bars. After completing the mineralization, the clear solution was cooled to room temperature, transferred to 50 mL graduated flasks, and filled with deionized water (ELGA Pure Lab Classic, High Wycombe, UK).

The parameters of the ICP-OES apparatus were the following: RF generator power 1150 W, RF generator frequency 27.12 MHz, coolant gas flow rate 16 L·min^−1^, carrier gas flow rate 0.65 L·min^−1^, auxiliary gas flow rate 0.4 L·min^−1^, maximum integration time 15 s, pump rate 50 rpm, viewing configuration–axial, replicates–three, flush time 20 s.

Multi-element stock solutions from Inorganic Ventures (USA) were used for the calibration of element concentrations: (1) CCS-4 for Al, As, Ba, Be, Bi, Ca, Cs, Ga, In, K, Li, Mg, Na, Rb, Se, and Sr in 7% (*v*/*v*) HNO_3_ (100 μg/mL); (2) CCS-5 for B, Ge, Hf, Mo, Nb, P, Re, S, Sb, Si, Sn, Te, Ti, W, and Zr in 7% (*v*/*v*) HNO_3_ and 1.2% (*v*/*v*) HF (100 μg/mL); (3) CCS-6 for Ag, Cd, Co, Cr, Cu, Fe, Hg, Mn, Ni, Pb, Tl, V, Zn in 7% (*v*/*v*) HNO_3_ (100 mg/L).

In this study, we applied the functional approach [19] for further data interpretation, and therefore determined the entire-body levels of the following bioelements which belonged to the following classes: (A) electrolytic bioelements (K, Na, Ca, and Mg) essential for the appropriate physiological potentials and osmotic balance; (B) enzymatic bioelements (V, Cr, Mo, Mn, Fe, Co, Ni, Cu, Zn, Sn, and Se which are the ions of metals), essential for catalytical functions of the cell metabolism; (C) exclusively toxic bioelements (Tl, Pb, Bi, Hg, As, Be, and Cd), which are lethal even when their tissue content is very low; (D) physiologically essential bioelements (Si, P, S, B, Sr, Ba, Ti, Al, Li, and Ag), which are important ingredients of the functional elements of the cells (among others proteins, lipids, carbohydrates, or nucleic acids). All symbols of these bioelements are compliant with the standards of the International Union of Pure and Applied Chemistry (IUPAC).

### 2.3. Statistics

Principal component analysis (PCA), comprising a correlation matrix, was applied. The effects of the IMD dose on the main variation components (PC1 and PC2) were determined based on the Cattell criterion. One-way ANOVA (factor–feeding group) along with the NIR post hoc test (*p* ≤ 0.05) were also applied to estimate the IMD dose effects and the significance of differences between the groups (α = 0.05). Statistica software, version 12.0, StatSoft Inc. USA, was used.

## 3. Results

During the feeding period, neither marked differences in the syrup and pollen consumption between the control, IMD5, and IMD200, nor any visible symptoms of the colony depopulation, were found.

PCA showed that the first major component (PC1) defined 57.56% of variability and described the cases of IMD use (Figure 1). Positive values of PC1 described bees that were not exposed to IMD (control; dotted ellipse). Negative PC1 values described worker bees exposed to 5 ppb of IMD (solid ellipse) or 200 ppb of IMD (dashed ellipse). The ellipses lie in different parts of the graph and form compact groups. This proves a severe influence of the diet type (control, IMD5, and IMD200) on PC1 and PC2 and, therefore, the severe influence of the IMD dose on the bioelement content in the worker bee bodies. All biominerals are plotted far from the center of the system, which points out to the fact that IMD strongly influenced their content in the worker bee bodies (Figure 2). As, Cr, Tl, B, Si, Sn, V, and PB (PC1 negative values) are most commonly found the bees exposed to IMD; Si best describes the dose of 5 ppb of IMD and Sn–200 ppb. Ba, Ag, Bi, and Sr are strongly and negatively correlated with the high dose of IMD. Pb is equally found in bees from the IMD5 and IMD200 groups. 

The levels of all the electrolytic bioelements in the worker bee bodies decreased due to IMD exposition proportionally to IMD dose, but the 5 ppb dose of IMD had a strong impact. (Figure 3). This corresponds with the PCA results (Figure 2) as all of the bioelements very highly influenced PC1 (their values were above + 0.8) and were also highly positively correlated with one another.

IMD decreased the levels of 8 out of the 11 enzymatic bioelements (Figure 4) but increased those of only 3 bioelements (SN, V, Cr). The impact of the feeding with addition of both 5 ppb and 200 ppb of IMD was similarly strong in about 50% of the bioelements, i.e., 5 ppb of IMD influenced the biomineral content in a really marked way. PCA showed (Figure 2) that all of these eight bioelements had high positive values of PC1 and were very highly positively correlated with one another. On the other hand, Sn, V, and Cr had negative PC1 values while being at the same time positively, but not closely, correlated with one another. All bioelements belonging to this class were very highly influenced by IMD, as values of PC1 or PC2 were higher than |0.8| for 10 out of 11 of them.

IMD decreased the levels of four but increased those of three out of the seven exclusively toxic elements (Figure 5) proportionally to its dose. Notably, IMD increased the levels of As, Pb, and Tl. This corresponds with PCA (Figure 2), which showed that As, PB, and Tl were positively correlated with one another but negatively with the remaining exclusively toxic bioelements, so these two bioelement sets influenced PC1 in a completely different way. Moreover, IMD had the weakest effect on this class of bioelements as the PC1 and PC2 values of five out of all seven bioelements ranged from −0.79 to 0.79.

IMD decreased the levels of 6 of out of the 10 physiologically essential bioelements (Figure 6) but increased it only in the case of B. The impact of IMD was strong in IMD5, and in almost 50% of cases, bioelement concentrations did not differ between IMD5 and IMD200. PCA showed (Figure 2) that all of those six bioelements were closely correlated with one another and reached high positive values of PC1. On the other hand, they were highly negatively correlated with B, which in turn, had negative values for PC1 and PC2. Interestingly, the IMD5 diet increased the Ba and Aa content, whereas the IMD200 diet decreased the Ba and Aa levels. Conversely, Si content was decreased with the IMD5 diet but increased with the IMD200 diet. PCA showed (Figure 2) that these three bioelements very highly influenced PC2, with Ba and Ag influencing it positively and Si influencing its negative values. Consequently, Ba and Ag were very closely positively correlated with each other but highly negatively correlated with Si. Only 1 out of the 10 bioelements of this class had PC1 or PC2 values lower than |0.8|, so IMD influenced them strongly.

Generally, the values of PC1 or PC2 were higher than |0.8| for 26 out of the 32 bioelements. IMD had the smallest impact on the exclusively toxic bioelements. Body levels of 22 out of the 32 bioelements were decreased with IMD, whereas body values of only 7 bioelements were increased with it. As regards the remaining three bioelements, marked IMD/dose × bioelement content interactions were noticed.

## 4. Discussion

The molecular formula of IMD (C_9_H_10_C_l_N_5_O_2_) did not contain any bioelements examined in this study, thus the addition of IMD to the diet of our bees, particularly in the case of IMD200, could not increase the level of any bioelement in their bodies by means of “direct addition”. In this study, we did not analyze IMD content in the worker bee bodies. However, we found [14] that its level was 0.35 ± 0.24 ng/bee (mean ± SD) in the corpses of about 100 worker bees from each colony, but only in the case of IMD200, when the same IMD diets and feeding protocols were used in our previous research.

### 4.1. Effects of Exposure to IMD on Honey Bee Bioelement Balance

We have revealed that the exposure to IMD unexpectedly and severely decreased K, Na, Ca, and Mg content in the worker bee bodies. Notably, even sublethal doses of IMD (5 ppb in IMD5), which proved to be field-relevant during the other studies [5,9] (after crops were treated with IMD), markedly decreased the levels of those electrolytic bioelements in our bees. All of them are important for the proper physiological potentials and osmotic conditions [19], maintaining water and constant pH in honey bee tissues, the electrical and osmotic status of apian cells, and also for the neutralization of lactic acid during the anaerobic breakdown of glucose in the flight muscles. Furthermore, high levels of K and Mg are necessary for proper insect development and growth [20,21], especially as Mg is a co-factor for many metabolic pathways. Decreased K content, in turn, appeared to be harmful especially to the forager bees, as K is necessary for the flight muscle activity. Ca appeared essential for signaling, particularly for nerve and muscle functions [20]. IMD impairs the effect of stimulus transmission in the insect nervous system and blocks nicotinic receptors, which may cause the accumulation of acetylcholine and in turn results in the paralysis or death of the insect. It may be related directly or indirectly to an impaired impulse transmission resulting from disturbances in Na^+^ and K^+^ titers (axons; membrane polarization) and an impaired functioning of neurotransmitters due to a disturbance of calcium channels (Ca^2+^) in the presynaptic membranes. Notably, senescence leads to a decrease in Ca, Na, and Mg content in apian bodies [25]. Summing up, since K+, Na+ Ca^2+^ and Mg^2+^ cations are considered to be crucial for honey bee fitness and health, and our study specifically showed that IMD caused a serious deficiency of these bioelement cations even in the doses, it can be concluded that IMD might exert a huge destructive impact on many honey bee metabolic pathways precisely in this way. This extends the contemporary knowledge about the potential mechanisms of behavioral, physiological, and anatomical handicapping of honey bees with neonicotinoids.

IMD decreased the levels of such enzymatic bioelements as Mo, Mn, Fe, Co, Ni, Cu, Zn, and Se in our bees. Similar to the electrolytic bioelements, the effect of the sublethal field-relevant dose of IMD was substantial for most of them. Most of the enzymatic bioelements form metal complexes and are important for catalytic functions in cell metabolism [19]. Consequently, the decreases in their levels caused by IMD, might impair many functions of the organism. For instance, Se and Zn are necessary for proper body maintenance, including immunity, and Cu and Zn, as well as the electrolytic bioelement Ca, are necessary for the anti-inflammatory response [20]. Furthermore, Cu has antimicrobial activity [26] and may protect lipids from peroxidation [27]. High Fe (and Mg) content is more important for the forager bees as Fe is necessary for their orientation with the Earth’s magnetic field. Moreover, Fe is an important component of cytochrome enzymes [28] and its content in bee bodies falls with age [25]. Mn, in turn, which is important for brain and muscle performance, is involved in the metabolism of such compounds as fats, sugars, and proteins as well as being important for the appropriate shaping of chitin. It is also a key element for antioxidant enzymes, which corresponds with our previous findings that IMD impairs antioxidative barriers in honey bees [15]. Ni is involved in many crucial functions of the organism as a cofactor for glyoxalase I, acireductone dioxygenase, superoxide dismutase, [NiFe]-hydrogenase, acetyl-coenzyme A synthase/decarbonylase, methyl-coenzyme M reductase, carbon monoxide dehydrogenase, and lactate racemase [29]. Consequently, when exposed to IMD, our bees faced bioelement deficiency with regard to the majority of enzymatic bioelements, which could result in disorders concerning many physiological processes. On the other hand, we have observed increased levels of such enzymatic bioelements as Sn, V, and Cr in the bodies of our bees due to IMD feeding. The excess of V and Cr could be toxic, but Cr is necessary for bee immunity and important for glucose metabolism [1,20]. However, an increase in the levels of Sn, V, and Cr accompanied by a decrease in the levels of the remaining enzymatic bioelements may result in a harmful stoichiometric imbalance between them [22]. This corresponds with the suggestions [21] that poorly balanced mineral diet may handicap physiological functions of bees. Summing up, IMD had an unexpectedly severe negative impact on enzymatic bioelements in our bees, which expands our suggestion that IMD may exert a destructive impact on honey bee physiology, first of all through bioelement deficiency but also through an impairment of the bioelement balance.

IMD did not affect all exclusively toxic elements as clearly as it did the remaining bioelement classes in our bees since it decreased the levels of four of the elements but increased those of three out of the seven of them. Nevertheless, the impact of the sublethal IMD dose was evident in this case as well. Particularly, the increases in Tl, Pb, and As content seem to be alarming as the elements can be very harmful even in trace amounts [19,20]. Furthermore, an excess of As impinges on the metabolism of Ca and Zn, damages the nervous system [30], and leads to oxidative stress. Pb can handicap proteins, cell membranes, reproduction, and the performance of numerous enzymes and can interfere with the proper metabolic functioning of Fe, Ca, Cu, Mg, and Zn and therefore interfere with the proper functioning of cells [20,30]. Consequently, IMD may interfere with honey bee physiology also by increasing the toxicity of some metals. Additionally, exposure to it may result in harmful stoichiometric biomineral imbalances [22]. Brodschneider and Crailsheim [21] also indicate that poor nutritional balance may additionally intensify the effects of many stressors, including pesticides, and accelerate honey bee colony losses.

IMD surprisingly and significantly decreased the body levels of the majority of physiologically essential elements, such as P, S, Ti, Al, Li, and Sr, but increased only the B content in the bodies of our bees. Interestingly and unlike the remaining bioelements, the interactions between the IMD dose and the bioelement content were observed among Si, Ba, and Ag; i.e., the level of a given bioelement in IMD5 was higher than in IMD200, or even in the control, or vice versa. Similar responses to IMD exposure were observed in the activity of certain proteolytic and antioxidant enzymes in our previous studies [14,15]. Notably, similar to the remaining bioelements, the impact of IMD was significant even when its doses were sublethal. P is the key element of the organism energy carriers [20], which is in line with the observation that IMD reduced oxygen consumption, impaired mitochondrial functions, and intensified glycolysis [3]. In turn, both Al, B, and Si deficiency and an excess of B and Si could accelerate senescence [22,31]. On the other hand, B is associated with coenzyme A, riboflavin, vitamin B6, vitamin B12, and nicotinamide adenine dinucleotide [32]. Summarizing, the imbalance and stoichiometric mismatch [22] between B, Si, Ba, and Ag accompanied with severe P, S, Ti, Al, Li, and Sr deficiencies resulting from the exposure of honey bees to IMD, which has been revealed in this study, could handicap many functions of the bees. Consequently, this might accelerate colony depopulation.

### 4.2. Interactions between Bee Exposure to IMD and Other Harmful External Factors

Our former studies [24] revealed that overwintering stress decreased the honey bee body levels of Ca, K, and Mg (electrolytic bioelements), Al, Cu, and P (physiologically essential bioelements), and Cr, Se and Zn (enzymatic bioelements) in temperate zones, showing that these bioelements might play essential functions in bee overwintering. In particular, the Ni content was reduced 4.8 times. Ilijević et al. [16] also reported lower levels of bioelements important for bees during winter and spring. It can be suggested in this context that bioelement shortages accompanied with bioelement imbalance caused by IMD, which we revealed in the present study, may intensify severe winter bioelement deficiencies and stochiometric mismatch. Taking into account the results of our current research presented here, we may suggest for the first time that the winter colony losses can be increased by the negative nutritional side effects of IMD, even in sublethal, field-realistic doses. This expands our knowledge about the reasons for the harmfulness of neonicotinoids and provides new tips for bee health prophylaxis.

Honey bees are exposed to many hazards nowadays, and negative synergistic interactions may occur between these threats, which also concerns insecticides [2,33]. However, the mechanisms of these interactions are still poorly known. Refs. [7,11] pointed out many ways in which the pesticides, including IMD, might decrease honey bee tolerance to such pathogens as *Varroa destructor* and *Nosema ceranae*. However, there was no mention about bioelements in this context. Our former study [24] revealed that *N. ceranae* specifically decreased the levels of Al, B, Fe, Si, K, Mg, Na, Cr, Mn, and Ni, i.e., those bioelements that are also connected with honey bee vitality and health. Compared to the bioelement deficiency and imbalance caused by IMD and revealed in this study, we can suggest that the synergic effects of N. ceranae and IMD could be proposed as an additional reason for decreasing bee tolerance to nosemosis, which has not been described to date. Particularly, the deficits of Si or B may increase the susceptibility to pathogens [17]. It is worth emphasizing that this can occur particularly when honey bees are facing low pollen supply, which is common in contemporary anthropogenic ecosystems [23,34].

### 4.3. Further Studies and Perspectives for Preventive Honey Bee Healthcare

Our results have shown that IMD causes unexpectedly severe bioelement deficiencies and imbalance in honey bee body tissues. This justifies further research on the mechanisms of this yet poorly explored side effect of neonicotinoids. Particularly, there are deficiencies in bioelements caused by the dearth of pollen, poor pollen biodiversity [22], and/or dearth of the natural astatic water pools in the contemporary agricultural ecosystems. Such research is of interest for nutritional ecology [23], but first of all, for contemporary amateur, semi-commercial, and commercial beekeeping that struggles with declining honey bee health status [34]. Unfortunately, neonicotinoids are widely present in the majority of the agroecosystems [5], particularly in field-relevant, sublethal doses which, as we have shown, are also harmful by disturbing the bioelement balance. As we have revealed that IMD decreased the body levels of 69% of the bioelements while increasing the levels of only 22% of them, the question arises how apiculture should face this negative combined effect of neonicotinoids and anthropogenic nutritional deficiencies. Consequently, we recommend developing methods of supplementing the honey bee diet and the natural bee food resources with bioelements when pesticides are present. This coincides with opinions of Manning [35] and Bonoan et al. [23], that there is a need for developing a supplemented diet for bees, as well as the opinions of Brodschneider and Crailsheim [21], that K and Mg have to be commonly used as supplements in bee nutrition. Strikingly, it was just the levels of these elements that were severely reduced in our bees as a result of exposition to IMD. Our efforts should concentrate on pollen supplements or on any other bioelement supplementation. The studies and applications of the artificially made so-called “dirty water” or mineral supplements for watering honey bees (compare [20,23]) should be also developed.

### 4.4. Additional Comments

IMD can impair the foraging behavior, homing success, navigation performance, and social communication of honey bees [36]. It reduced the foraging activity, disturbed the olfactory learnt discrimination tasks [37,38], decreased the rate of revisiting food resources, delayed forager return flights [39], and altered navigation memory [40]. Furthermore, bumblebees exposed to IMD collected less pollen than the unexposed ones as they learned less efficiently the reward value of flowers and therefore were less efficient at nectar foraging [41]. Juho et al. [42] confirmed that bumblebees exposed to neonicotinoid at concentrations up to 1 ppb changed their foraging behavior, showing a reduction in foraging motivation, and consequently, poor hoarding of the sucrose solution. The authors concluded that neonicotinoids may reshape interactions between pollinators and plants. Additionally, more bumblebees in the colonies exposed to sublethal, field-realistic levels of neonicotinoids turned to foraging more frequently, their colonies collected more pollen, and importantly, they visited more *Lotus corniculatus* flowers than the controls, which, in turn, visited more *Trifolium repens* flowers. This shows that bees may alter their floral preferences and hoarding efficiency when exposed to neonicotinoids [43]. On the other hand, our research has shown that chronic exposure to IMD supplied to the within-hive feeders led to unexpectedly severe bioelement deficiencies and imbalance in honey bees even when administered with sublethal field-realistic doses of 5 ppb (compare [2,5,9,41]). Taking into account the findings mentioned above [36,37,38,39,40,41,42], doubts may arise whether the differences between the bioelement contents in our controls versus IMD5 and IMD200 resulted from IMD-related disturbances of the apian metabolism, because IMD could also alter the foraging preferences of the IMD5 and IMD200 bees, which consequently could consume food with different bioelement contents. One way or another, it was IMD that was the triggering factor. Therefore, the only question is whether IMD causes disturbances in the honey bee body bioelement content by impairing apian metabolism or rather through altering the apian foraging preferences. This is also an interesting issue for further research. Notably, changes in honey bee foraging preferences may also depend on whether the food resources containing IMD were situated outside or inside the hives [36,37,38,39,40,41,42]. However, since there were no floral resources available during our experiment, all the colonies (12 colonies within each of the 3 feeding groups) were genetically and structurally similar and obtained the same unified food throughout the entire experiment. Since the effects of IMD exposure appeared to be unexpectedly significant and steady, we believe the differences between the controls versus IMD5 and IMD200 could not be due only to inter-colony variance. To answer the question whether IMD impairs the bee-body bioelement balance by altering honey bee metabolism or by altering the honey bee foraging behavior, or by both factors, further experiments in environments in which the natural food resources are available, or unavailable, should be performed. This corresponds with the suggestion of Felicity et al. [44] that “bumblebees did not preferentially visit floral stimuli previously paired with a neonicotinoid-containing solution, which also points out the need for further studies on mechanisms of neonicotinoid-driven foraging preferences in different bee species”. In our former experiments, in which the same experimental design and the same bee foods were applied, we found the following. The IMD levels in the syrup and pastry were approximately 4.2 ppb and 196 ppb in IMD5 and IMD200, respectively, during the 3 months following food preparation [15]. The IMD residuals were found to be 0.35 ± 0.24 ng/bee in the bodies of about 100 workers at the age of 1–10 days sampled from each colony, but it was only in the IMD-200 group [14]. The IMD residuals amounted 0.0 ng/bee in controls and IMD5 and 0.48 ± 0.38 ng/bee in IMD200, when the 100 ten-day-old workers were assayed within each colony [15]. The IMD concentration in the comb storage was 0.0 ppb in the controls, 4.1 ± 0.51 ppb in IMD5, and 111.7 ± 56.33 ppb in the IMD-200 group [15]. Therefore, based on these results, it can be concluded that every bee sampled for determining the body bioelement content in this study had consumed IMD as well.

## 5. Conclusions

This study expands the knowledge about mechanisms of pesticide harmfulness to honey bee health and, consequently, when compared with the findings of other researchers, also the ways IMD may handicap behavioral, physiological, and health traits in honey bees, pointing to bioelements as important metabolic components unexpectedly severely disturbed by IMD. This is a new observation. We believe that such adverse side effects of neonicotinoids should be especially strong in the contemporary agroecosystems, in which natural bioelement resources for pollinators are scarce. Since bioelement deficiency may also be linked to apian immunity and resistance, we suggest that the IMD-dependent disturbances in the bee body bioelement content could be proposed as a yet another new mechanism of decreasing their tolerance to invasive pathogens and, importantly, increasing the colony winter loses, which has not been considered to date.

Further studies and practical apiary applications should concentrate on the bioelement supplementation of the honey bee diet, not only as a way of protecting them when they face malnutrition in anthropogenic ecosystems but also as a new way of compensating the negative influence of neonicotinoids. Our results suggest that the mineral supplementation of the honey bee diet, including the commercial winter bee-food or the water resources for bees, including “dirty water”, is a promising way of protecting apian health when the insects are exposed to pesticides. These applications can be potentially simple, cheap, and not very time-consuming.

## Figures and Tables

**Figure 1 animals-13-00615-f001:**
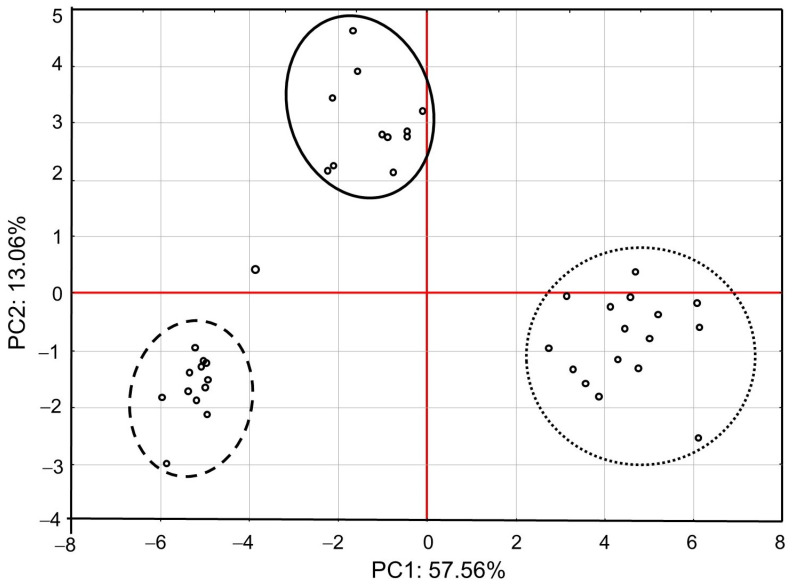
Principal component analysis (PCA) in honey bees exposed and unexposed to imidacloprid: two-dimensional projection of the imidacloprid dose in the principal components PC1 and PC2. Explanations: doted-line ellipse = bees were fed a diet without imidacloprid-control group; solid-line ellipse = bees were fed a diet containing a 5 ppb addition of imidacloprid; dashed-line ellipse = bees were fed a diet with a 200 ppb addition of imidacloprid.

**Figure 2 animals-13-00615-f002:**
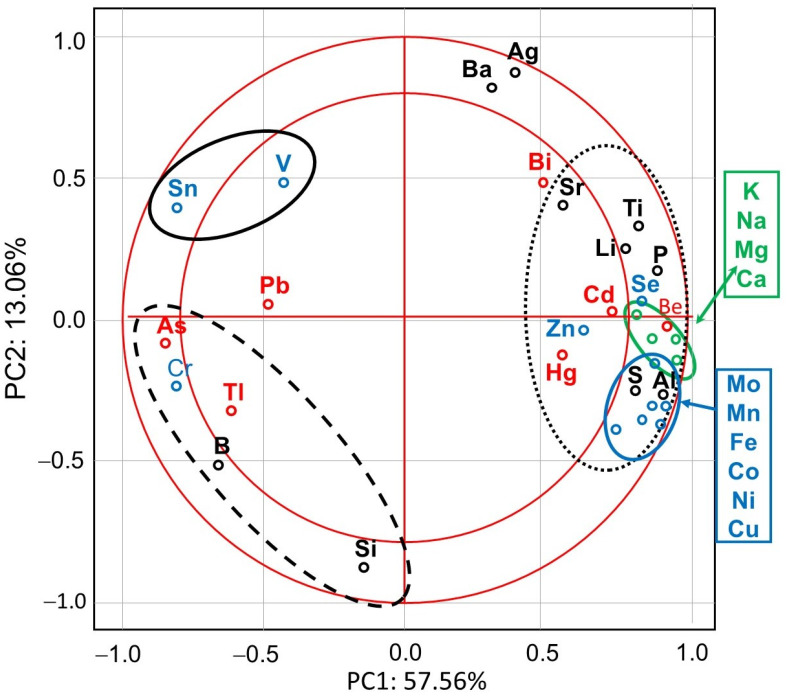
Principal component analysis (PCA) in honey bees exposed and unexposed to imidacloprid: two-dimensional projection of the variables (biominerals) in the principal components PC1 and PC2. Explanations: doted-line ellipse = bees were fed a diet without imidacloprid-control group; solid-line ellipse = bees were fed a diet containing a 5 ppb addition of imidacloprid; dashed-line ellipse = bees were fed a diet with a 200 ppb addition of imidacloprid.

**Figure 3 animals-13-00615-f003:**
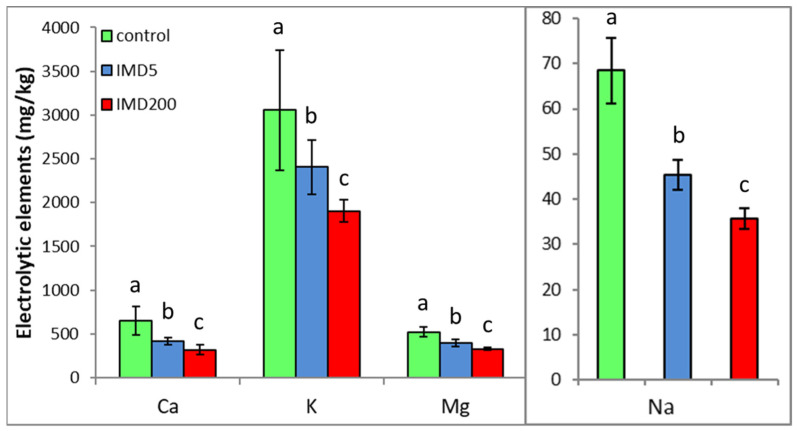
Body levels (mean ± SD) of the electrolytic bioelements in honey bee workers exposed and unexposed to imidacloprid. Explanations: control = bees were fed a diet without imidacloprid; IMD5 = bees were fed a diet containing a 5 ppb addition of imidacloprid; IMD200 = bees were fed a diet with a 200 ppb addition of imidacloprid. Different lowercase letters = differences between the means nested within each bioelement are significant (ANOVA + LSD; *p* < 0.05).

**Figure 4 animals-13-00615-f004:**
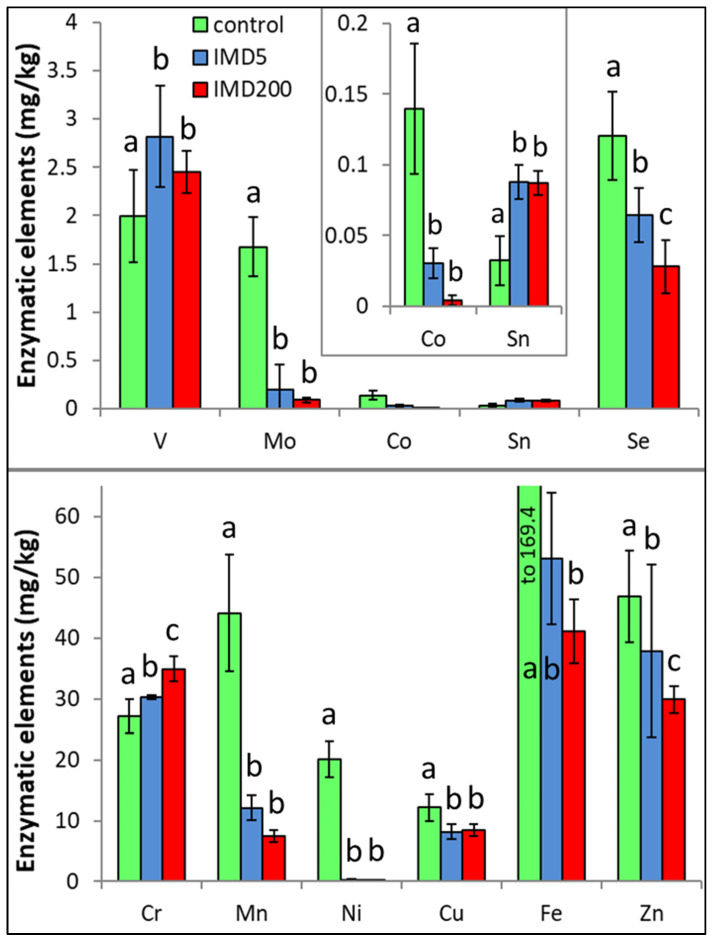
Body levels (mean ± SD) of the enzymatic bioelements in honey bee workers exposed and unexposed to imidacloprid. Explanations: control = the bees were fed a diet without imidacloprid; IMD5 = the bees were fed a diet with a 5 ppb addition of imidacloprid; IMD200 = the bees were fed a diet with a 200 ppb addition of imidacloprid. Different lowercase letters = differences between the means nested within each bioelement are significant (ANOVA + LSD; *p* < 0.05).

**Figure 5 animals-13-00615-f005:**
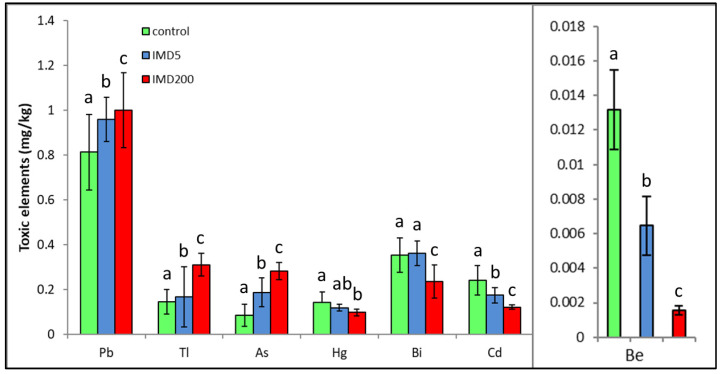
Body levels (mean ± SD) of the exclusively toxic bioelements in honey bee workers exposed and unexposed to imidacloprid. Explanations: control = the bees were fed a diet without imidacloprid; IMD5 = the bees were fed a diet with a 5 ppb addition of imidacloprid; IMD200 = the bees were fed a diet with a 200 ppb addition of imidacloprid. Different lowercase letters = differences between the means nested within each bioelement are significant (ANOVA + LSD; *p* < 0.05).

**Figure 6 animals-13-00615-f006:**
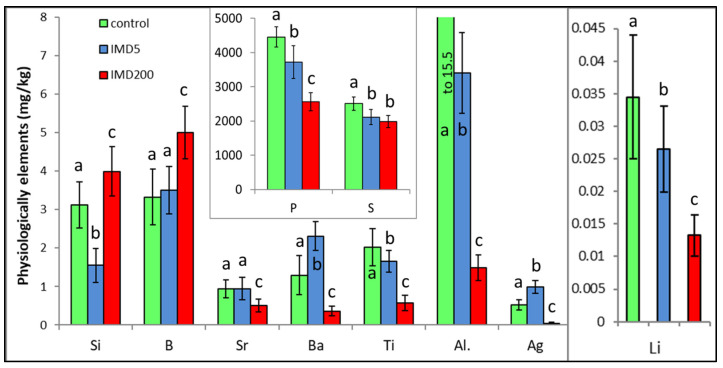
Body levels (mean ± SD) of the physiologically essential bioelements in honey bee workers exposed and unexposed to imidacloprid. Explanations: control = the bees were fed a diet without imidacloprid; IMD5 = the bees were fed a diet with a 5 ppb addition of imidacloprid; IMD200 = the bees were fed a diet with a 200 ppb addition of imidacloprid. Different lowercase letters = differences between the means nested within each bioelement are significant (ANOVA + LSD; *p* < 0.05).

## Data Availability

The datasets generated and analyzed during the current study are available from the corresponding author upon request.

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
