# Peer review of "Imidacloprid Pesticide Causes Unexpectedly Severe Bioelement Deficiencies and Imbalance in Honey Bees Even at Sublethal Doses"

_animals, 2023, doi:10.3390/ani13040615_

Round 1
Reviewer 1 Report
The authors suggested that even residual level of imidacloprid (IMD) can disturb the bioelement balance of honey bee. They examined the bioelement of honey bee with 5 ppb and 200 ppb IMD treatment and confirmed that 5 ppb IMD can cause severe impact on the bioelement constitution.
Major comment
As IMD was applied in different colonies, the colony differences become the greatest variance in this study. How was the colony before IMD treatment? From the current M&M sections, we cannot conclude that those bioelement differences were generated by imidacloprid rather than colony differences before the treatment. A pretest to examine the bioelement constitution before the treatment is required.
Minor
· Honey bee is the scientific term, please revise the “honeybee” to “honey bee” throughout the manuscript.
· I was wondering if the author tested the residue level of IMD in samples? It will be another way to confirm the correlation between the intake of IMD and the changing of bioelement.
· Line 120, fifteen worker bees were collected for biomineral composition analysis. So 15 bees per sample/replicate? How many samples/replicates per treatment? It is not described in the manuscript.
· For the reference, make sure the format is consistent.
Ref 1, journal is provided in a full name.
Ref 25 and 26 show no DOI.
Author Response
My response is in the attached file

Reviewer 2 Report
Pesticides impair honeybee health in many ways. the study investigated impact of the imidacloprid pesticide (IMD) on the bioelement body content in honeybees in the field experiments. the results revealed for the first time that IMD causes unexpectedly severe bioelement deficiencies in 69% of bioelements and imbalances contents of the remaining ones even in the sublethal, residual doses. The increase in three toxic bioelements was alarming. the authors suggested a new physiological mechanism; how the nicotinoids may interfere the honeybee health status. This also give directions for further research, pointing to the bioelement supplementation of the diet as an important element of honeybee preventive health care when the bee-farms or amateur apiaries are located in agrocenosis exposed to pesticides.I suggest some minor revisions of the text.
1. The title of the manuscript is too long.
2. Figure legends, is it the style requirement of the journal? Each figure legend start with explanations. Is it necessary?
3. Informed Consent Statement, delete the quotation mark at the end of the sentence.
Author Response
My response is in the attached file

Reviewer 3 Report
Please check the attachment

Author Response
My response is in the attached file

Round 2
Reviewer 1 Report
Major comment
As IMD was applied in different colonies, the colony differences become the greatest variance in this study. How was the colony before IMD treatment? From the current M&M sections, we cannot conclude that those bioelement differences were generated by imidacloprid rather than colony differences before the treatment. A pretest to examine the bioelement constitution before the treatment is required.
Response 1: The major revision of the M&M section has been done.
The completed text was as follows:
“…… Procedures described by [14, 15] were applied as follows. The mini-plus hives (six frames of 251×159 mm placed in one super) were populated with Apis mellifera carnica worker bees to establish experimental colonies of similar strength and structure. The worker bees originated from a few source colonies belonging to the same commercial stock and both the experimental and the source colonies had been maintained in the same location for two years. Furthermore, the experimental colonies were headed by egg laying, one-year-old sister-queens. Each of them was instrumentally inseminated with the semen of drones from the same mother-queen. Then, these experimental colonies, which were genetically and structurally similar and homogeneous, were used to set up three feeding groups. Each group consisted of 12 colonies. Bees in the control group were fed with pure sugar-water syrup (5 : 3 w/w). In the IMD5 group, the syrup contained 5 ppb of IMD (Bayer Health Care AG, Leverkusen, Germany) and in the IMD200 – 200 ppb of IMD. The colonies were also supplied with commercial bee food (API-Fortune HF 1575, Bollène, France) with the addition of ground, mingled, multifloral pollen containing 0 ppb, 5 ppb, and 200 ppb of IMD in the control, IMD5, and IMD200 groups, respectively. In this case the pollen was a unified source of biominerals. Importantly, the natural bee food containing potential bioelement resources was not available at that location during the study and the IMD was not used in the region. Consequently, the only source of IMD for the experimental colonies was the syrup and the commercial bee food, as well as the bioelement contents of diets of the experimental colonies ought to be similar to each other……….
I hope that is clear now that the diet (experimental factor) variance was evident and the colony variance has been minimized. Importantly, this has been also clearly shown by PCA. And that's why this statistical procedure has been made by us.
IMD treatment can alter the foraging behavior of honey bee, even flower preference (see 0.1007/s10646-015-1537-2). From the experiment, the whole bee was used for PCA analysis without removing the gut, thus the pollens / nectar from the plants were also included in the samples. Pollens / nectar of different species of flower may contain different biominerals and consequently generate great effect on PCA results. Using current protocol cannot exclude whether the differences were from the food (food resources or food amount) or from IMD treatment. The authors clarified that there was no floral resource available during the experiment, Line 120, but bees can travel for more than 10 km for foraging and thus can always be unpredictable events in the field. If an additional experiment is not applicable right now, I would suggest making another statement to include the potential effects from food resources and behavioral changes. It will be great, in the future, to test every colony before the experiment, to exclude those potential defects.
Minor comment
I was wondering if the author tested the residue level of IMD in samples? It will be another way to confirm the correlation between the intake of IMD and the changing of bioelement.
Response 3: No he didn’t. In fact we tested the 5 ppb level of IMD in the administered bee food.
This level was similar to the level, which had been found in the field conditions (after crops were treated with IMD) and was considered residual by other authors (residual - field relevant). We have used the proper references in ms. However it seems that there is a certain misunderstanding. Consequently we had added the proper explanations throughout the text – if we have understood this remark properly.
It is good to provide more detail in the manuscript. For the “samples” in my question, it refers to the honey bee sample but not the bee food. This is an “exposure” experiment, it is not easy to confirm if every bee collected had consumed IMD. If the authors can detect IMD residue in the IMD-treated honey bee sample, then the evidence will be stronger.
Author Response
Dear Reviewer 1
The detailed answer is in the attached files
Thank you for constructive comments
Best regards
